# Total Cholesterol Determination Accuracy in Dried Blood Spots

**DOI:** 10.3390/diagnostics14171906

**Published:** 2024-08-29

**Authors:** Elena Bonet Estruch, María J. López-Lara, Eva N. Gutiérrez-Cortizo, Miguel A. Castaño López, Pedro Mata, Manuel J. Romero-Jiménez

**Affiliations:** 1Clinical Chemistry and Laboratory Medicine Department, Infanta Elena Hospital, 21007 Huelva, Spain; mar532@gmail.com; 2Fundación Andaluza Beturia para la Investigación en Salud (FABIS), Infanta Elena Hospital, 21007 Huelva, Spain; investigoinfantaelena@fabis.org; 3Internal Medicine Department, Lipid and Vascular Risk Unit, Infanta Elena Hospital, 21007 Huelva, Spain; nadejda.gutierrez@gmail.com (E.N.G.-C.); manujromeroj@gmail.com (M.J.R.-J.); 4Internal Medicine, Spanish Familial Hypercholesterolemia Foundation, 28010 Madrid, Spain; pmata@colesterolfamiliar.org

**Keywords:** total blood cholesterol, dried blood spots, familial hypercholesterolemia, newborn screening

## Abstract

Background Detecting total cholesterol in dried blood spots could aid in identifying individuals with a high likelihood of familial hypercholesterolemia and could be used as a screening measure. This study aims to assess the diagnostic accuracy of dried blood spots on Whatman 903 paper cards using a manual enzymatic technique. Methods: A total of 394 samples were collected as serum and dried blood spots were compared. Cholesterol was determined in serum using the automated reference method, while cholesterol on paper was measured using a manual enzymatic method. Within- and between-day diagnostic variability were analyzed. The correlation between both methods was assessed using Passing–Bablok regression and Bland–Altman plot. Internal validation of our correlation formula was performed on 149 samples, along with external validation of the formula proposed by Corso et al. Results: The within- and between-day coefficient of variation was found to be lower than 10.14% and 14.09%, respectively. Passing–Bablok regression indicated a precision of 0.803 and an accuracy of 0.96. Internal validation precision was measured at 0.716. The resulting positive and negative predicted values were 0.77 and 0.92, respectively, vs. 0.46 and 0.96 from the external formula. Conclusions: Total cholesterol analysis in dried blood spots demonstrates high precision and reproducibility. This method reliably enables the incorporation of this biological marker into neonatal screening for familial hypercholesterolemia detection.

## 1. Introduction

Familial hypercholesterolemia (FH) stands as a genetic disorder caused by mutations in cholesterol metabolism genes, primarily low-density lipoprotein receptor gene (*LDLR*), and to a lesser extent, apolipoprotein B gene (*APOB*) and proprotein convertase subtilisin-kexin 9 gene (*PCSK9*) [1,2]. Its prevalence is estimated at 1 in every 250 individuals, impacting over 25 million people globally [3]. This disease constitutes the most prevalent genetic predisposition to premature atherosclerotic cardiovascular disease (ASCVD), with affected patients exhibiting a 3 to 13 times higher risk of ASCVD compared to the general population. Their life expectancy may be shortened by 20–30 years due to persistent exposure to elevated blood cholesterol levels since birth [4]. Treatment with lipid-lowering therapies initiated at an early age can ameliorate cardiovascular morbidity and mortality up to tenfold [1]. However, presently, nine out of ten individuals born worldwide with FH remain undiagnosed [5]. A recent systematic review by Beath Jhan et al. [6] outlined various strategies for FH diagnosis, concluding that the most cost-effective approach is universal screening in childhood alongside diagnosing first-line relatives through reverse cascade. In an economic evaluation using a decision tree analysis, the results showed that a national program for FH based on molecular testing is a cost-effective diagnostic and management strategy that supports government expenditure aimed at preventing coronary artery disease in FH patients [7]. Despite such recommendations, FH diagnosis has not increased substantially to date.

Newborn metabolic and endocrine disease screening programs constitute a successful public health initiative attaining a coverage exceeding 99%. Level 1 genomic alterations are defined by the Centers for Disease Control and Prevention Office of Public Health Genomics, as those that can be detected by genetic testing and their detection can have a positive impact on public health through early interventions to reduce their morbidity and mortality. FH is considered one of these level 1 genomic alterations [8].

Several authors have investigated biomarkers that evaluate the probability of presenting FH. Held et al. [9,10] have contributed studies regarding the validity of determining total cholesterol, low-density lipoprotein cholesterol (LDLc), and apolipoprotein B in dried blood spots (DBS). Corso et al. [11] devised and validated a method for determining cholesterol in DBS.

In our Clinical Analysis and Vascular Risk Laboratory at the Infanta Elena Hospital in Huelva we have enhanced a technique with specific modifications based on literature models [9,10,11]. The aim of our study is to ascertain the precision and stability of cholesterol levels using our methodological approach, correlate cholesterol levels in DBS with the reference method in serum, and conduct an internal validation of the regression formula, in order to carry out a pilot project to implement national FH screening in Spain.

## 2. Methods

### 2.1. Study Design

This study adopts a prospective cross-sectional design, comparing a series of samples using two distinct techniques for measuring cholesterol levels. The study adheres to the STARD 2015 guidelines [12] for diagnostic accuracy studies. The protocol of this investigation (Protocol code: CHOLESTEROL BDS-2023-01, titled “Accuracy of Blood Cholesterol Determination on Whatman 903 Paper (DBS Cholesterol Validation)”) was conducted in accordance with the Declaration of Helsinki and received approval from the Ethics Committee of the province of Huelva on 21 July 2023.

Participants were selected through consecutive non-probabilistic sampling. Individuals were eligible for the study if they were over 18 years of age, had a requisition for serum cholesterol determination from their attending physician, and consented to participate when visiting the Clinical Analysis laboratory of the Infanta Elena Hospital. Those who did not provide informed consent and individuals with insufficient or altered samples for both methods were excluded. Sample collection occurred during December 2023.

The measured variables included total cholesterol in DBS and serum.

### 2.2. Serum Total Cholesterol Method 

The gold-standard serum cholesterol method was enzymatic colorimetric determination with total cholesterol reagents manufactured by Roche Diagnostics SA in a COBAS PRO instrument (Roche Diagnostics International Ltd., Reinach, Switzerland, Sarstedt AG & Co). Both the reagent and equipment possess appropriate authorizations and *Conformité Européene* marking for cholesterol determination. Rigorous quality control measures are in place, including daily internal controls to verify operational integrity and result validity. Monthly external controls were conducted, comparing equipment performance with other national laboratories to ensure result accuracy.

### 2.3. Cholesterol Method in DBS

Cholesterol concentration in DBS was determined through a manual enzymatic colorimetric method employing Roche reagent (CHOL2 for Cobas equipment by Roche Diagnostics). For each determination, two disks of three-millimeters were extracted from each patient’s DBS card. The determination was conducted in duplicate, needing a total of four discs per patient. These discs were placed in a single well of a 96-well plate. To extract the blood, 125 µL of methanol (methanol-anhydrous, 99.8%, EMSURE^®^ ACS, ISO, Reag. Ph Eur) were added to each well. The plate was then covered with protective plastic and incubated at 37 °C for 30 min with gentle shaking, followed by 15 min without agitation. Subsequently, 100 µL of reagent and 50 µL of the extracted solution from the previous plate were added to another 96-well plate. The reaction was incubated at 37 °C for ten minutes, after which absorbance was measured at 492 nm using a microplate reader (AMR-100T, Hangzhou Allsheng Instruments Co., Ltd., Hangzhou, China ^®^).

In each plate, a calibration curve was established using five controls with known cholesterol levels ranging from 130 to 300 mg/dL. An internal control of known value was analyzed at the beginning and at the end of each plate. Absorbance levels for each patient were extrapolated to the calibration curve equation, thereby calculating cholesterol concentration in DBS. The median of all cholesterol values determined in each plate was computed and assessed in a Levey–Jennings chart, depicting the mean of all medians and standard deviation. Results were deemed acceptable if the median fell within the two standard deviations depicted in the chart.

The laboratory technician conducting DBS cholesterol determination was not privy to serum cholesterol values for each patient. Likewise, the technician performing serum cholesterol determination remained unaware of DBS cholesterol values.

### 2.4. Method Validation

For the analysis of diagnostic variability, three samples were selected for quality control (QC). Their serum cholesterol levels were 149, 220, and 283 mg/dL, respectively. These samples underwent an average of 25 replications on the same day (within-day) and 27 replications over different consecutive days (between-day).

### 2.5. Methods Comparison: SERUM/DBS

The precision assessment involved the determination of cholesterol in DBS as the index test. After obtaining informed consent, 394 patients were selected. Serum cholesterol values were collected and measured on the same day of blood collection using the COBAS PRO autoanalyzer at the Clinical Analysis Laboratory of the Infanta Elena Hospital. One tube of ethylenediaminetetraacetic acid (EDTA) was collected from each subject to prepare the DBS.

Figure 1 shows the flow chart of serum and DBS samples. Cholesterol values in DBS were determined in duplicate following the previously described methodology. Subsequently, a method comparison was conducted using Passing–Bablok regression, resulting in a regression line enabling the prediction of calculated cholesterol values in DBS. 

### 2.6. Internal Validation

For the internal validation of our regression formula, cholesterol was determined in DBS from 155 samples. The predicted cholesterol was then calculated based on the predictive formula. A Passing–Bablok regression compared the cholesterol results obtained using the predictor formula with those obtained using the serum reference method. Additionally, an external validation utilizing the formula proposed by Corso et al. [11] was conducted using the same samples. 

### 2.7. Statistical Analysis

Mean, standard deviation, and coefficient of variation were employed as statistical tools for method validation. To compare methods, a Bland–Altman plot was constructed to elucidate differences between cholesterol determination in DBS and serum. Data were analyzed using a paired non-parametric test (Wilcoxon test). A Passing–Bablok regression was conducted between cholesterol values measured in serum and DBS, resulting in a regression line. To validate this line, a box and whisker plot and another Passing–Bablok regression were used. Statistical differences in the cholesterol validation data were assessed using a Wilcoxon test. Cholesterol outlier values were identified using the Tukey method. Statistical analysis was performed using SPSS version 23 for data descriptions, and method comparison was accomplished by the R Statistical Software (v4.4.0; R Core Team 2024). The Bland–Altman plot was obtained from Bland–Altman Leh:Plots (Slightly Extended) R package (v0.3.1; 2015), and the Passing–Bablok plot from mcr (Method Comparison Regression) R package (v1.3.3; 2023). 

## 3. Results

Table 1 presents imprecision data, expressed as the coefficient of variation (CV) of DBS samples analyzed using the enzymatic method. Within-day imprecision was calculated based on three quality control samples with known values (149, 220, and 283 mg/dL), analyzed 23, 25, and 26 times, respectively. The CVs were 10.14%, 8.90%, and 8.11% for each respective level. For the between-day imprecision study, the same control samples were analyzed over 27 working days, resulting in CVs of 14.09% for QC 149 mg/dL, 10.57% for QC 220 mg/dL, and 13.29% for QC 283 mg/dL.

To compare methods, 394 patients were selected (Figure 1). Serum cholesterol levels were determined for all patients, along with DBS analysis. DBS samples were analyzed in duplicate. In cases where the difference between the two DBS measurements was <50 mg/dL (n = 333), the average cholesterol value was calculated. For cases where the difference exceeded 50 mg/dL (n = 61), a third and fourth determination were performed. Three subjects had insufficient samples for repeated analysis, and 16 exhibited differences exceeding 50 mg/dL between determinations, leading to their exclusion from the study. Consequently, the final sample size comprised 375 subjects with serum cholesterol determination and mean DBS values, which was reduced to 361 patients after identifying outliers using the Tukey test.

Furthermore, to conduct method comparison analysis, we decided to calculate the difference between serum cholesterol and DBS and discard values exceeding the 90th percentile or falling below the 10th percentile of the difference, to capture the central variability of data and provide a balanced representation of the studied population. The final sample comprised 289 subjects. Wilcoxon test was applied to analyze differences between both methods. Table 2 describes cholesterol results obtained using the serum reference method and the manual DBS method, indicating a mean serum cholesterol of 164 ± 41 mg/dL and a mean DBS cholesterol of 175.18 ± 44.10 mg/dL (*p* = 0.001).

Bland–Altman plot (Figure 2) was used to evaluate agreement between DBS and serum cholesterol determination, revealing an average bias of 11.5 mg/dL (solid dark-blue line) with a 95% confidence interval in green dot-dashed line of 8.40–14.60 mg/dL. The computed limits of agreement corresponding to ±1.96 SD are from −101.5 to 124.6 mg/dL (dashed gray lines) with a 95% confidence interval (light solid blue lines). The Pearson correlation coefficient between the averages and the differences is 0.80 (95% CI: 0.76–0.84). Pitman’s test used with related samples gives a result of *p* < 0.001 which means that the two distributions have different levels of dispersion.

We used the Passing–Bablok regression (Figure 3) to assess the agreement, accuracy, and precision of cholesterol measurement using DBS versus serum as the standard method. The Pearson correlation coefficient of 0.80 suggests a strong and consistent association between DBS and serum measurements. The bias correction factor (Cb) [13] value, which is a measure of accuracy, was 0.9604, suggesting high accuracy of the DBS method. The Passing–Bablok regression yielded an estimated intercept of 0.31 (95% CI: −13.44–+12.41) with corresponding estimated slope of 0.92 (95% CI: 0.85–0.99).

Subsequently, an internal validation of the method was conducted on 155 samples, processed in the same manner. Six samples were excluded due to variations > 50 mg/dL between repetitions. Using the previously obtained regression line, the cholesterol value for each subject was estimated from the resulting DBS cholesterol value. Comparing the results of serum cholesterol versus calculated cholesterol yielded a Passing–Bablok regression line with an estimated intercept of −43.24 [95% CI: −74.54–(−16.68)] with a corresponding estimated slope of 1.17 (95% CI: 1.03–1.35) (Figure 4).

We applied the regression line (y = 64.86 + 0.5217x) with x = estimated serum cholesterol and y = DBS cholesterol from Corso et al. [11] to the same validation population (n = 149) for external validation of the methodology. Non-parametric tests were conducted to determine significant differences between these data and the corresponding serum cholesterol value. Significant differences (*p* < 0.001) between serum–DBS, serum–Corso and DBS–Corso were noted (Figure 5).

In addition, using a total cholesterol cutoff value of 190 mg/dL, we observed that in DBS 76 of 91, cholesterol predicted values of the validation test were below 190 mg/dL and in accordance with the corresponding serum cholesterol values, and 51 of 58 were above 190 mg/dL in accordance with the serum values. Nevertheless, using Corso’s formula, we obtained 23 of 91 cholesterol values below 190 mg/dL and in accordance with the corresponding serum cholesterol values, and 57 of 58 above 190 mg/dL.

Applying our regression formula, a sensitivity value of 0.88 and a specificity value of 0.84 with a positive predictive value of 0.77 and a negative predictive value of 0.92 was found, whereas with Corso’s formula, a sensitivity of 0.98 and specificity of 0.25 had been obtained, with a positive predictive value of 0.46 and a negative predictive value of 0.96.

## 4. Discussion

Familial hypercholesterolemia (FH) remains underdiagnosed despite being recognized as a public health priority by the World Health Organization (WHO) in 1988 [5]. Approximately 450,000 children are born with FH annually worldwide, yet only 2.1% of adults with FH are diagnosed before the age of 18 using current diagnostic approaches [14].

The increase in cholesterol levels since birth presents an opportunity for early FH diagnosis. Early identification and cholesterol reduction can prevent premature ASCVD. Implementing universal screening for FH in childhood is a logical approach to close the gap between prevalence and detection. This approach is in line with the 2020 WHO–UNICEF–Lancet Commission Strategy [15], which emphasized the importance of early preventive interventions in childhood over corrective actions in adulthood. 

In this study, we developed and validated a method for extracting and analyzing total cholesterol from DBS. DBS analysis reduces the invasiveness of blood sampling and requires only a small sample volume, facilitating easy transportation to laboratories.

The precision results obtained were within acceptable limits (CV 10%) for total cholesterol determination. Cholesterol concentrations remained stable over 27 consecutive days of measurement. 

Sample preparation quality significantly influences both analysis and cholesterol extraction. The use of calibrated and certified filter paper cards is crucial for the analysis of blood biomarkers; in addition, the amount of blood that is deposited must be equal to or greater than 70 μL with homogeneous distribution to minimize variability. Each determination was accompanied by a calibration curve and internal controls analyzed under consistent conditions on the same day. Sample determination was performed in duplicate. Comparisons of daily medians on a Levey–Jenning chart was performed, excluding those that exceeded two standard deviations. The final result was accepted if it fulfilled all the quality conditions.

Comparisons between serum cholesterol and DBS determinations of concurrently obtained samples from research participants using the applied algorithm showed positive agreement. DBS cholesterol values were consistently higher, attributed to the presence of cholesterol in red blood cells. This finding is consistent with reports by Corso [11] and Held [9,10]. In our study, the observed difference was smaller (11 mg/dL) compared to the 30 mg/dL found in previous studies [10], possibly due to the 16% residual cholesterol present in the paper post-extraction, as described by Corso et al. [11] 

### 4.1. Study Limitations

Manual determination increases imprecision compared to automated analyzers, but currently, no auto-analyzers offer such technology. Standardizing sample collection with paper is challenging, requiring prior extraction and analyte dilution. 

### 4.2. Advantages

Universal neonatal screening uses DBS. The validation of cholesterol determination in this sample can be used as a screening test for the diagnosis of FH. The determination of cholesterol in DBS has been carried out with instrumentation already available in neonatal screening laboratories and the cost of the determination is equivalent to the cost of the reagent, facilitating adaptation in all screening laboratories for this disease.

### 4.3. Future

Further population studies with neonatal DBS samples are needed to establish cholesterol level cutoffs for confirmatory genetic testing. Evaluating cost-effectiveness and healthcare circuits will aid in expanding this procedure.

## 5. Conclusions

Cholesterol determination using DBS is a cost-effective, accessible, reproducible screening method for selecting individuals at risk of FH. Confirmatory genetic testing remains essential for diagnosis.

## Figures and Tables

**Figure 1 diagnostics-14-01906-f001:**
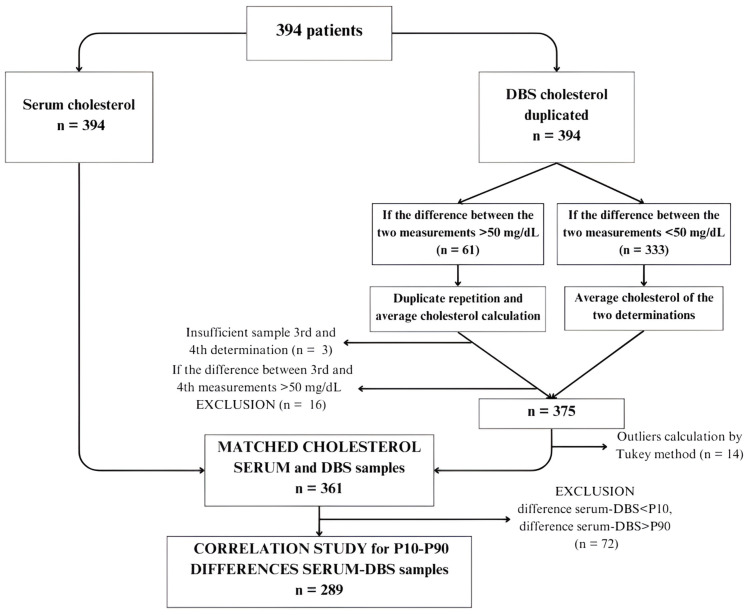
Flow chart of samples in the correlation study. (DBS: dried blood spots; P10: percentile 10; P90: percentile 90).

**Figure 2 diagnostics-14-01906-f002:**
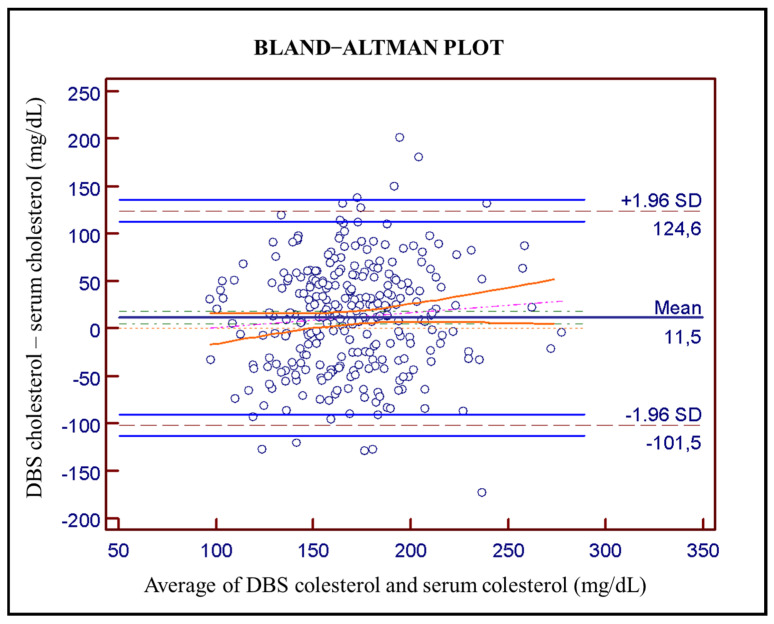
Bland–Altman plot between serum cholesterol levels and DBS cholesterol levels (n = 289; average 11.5 mg/dL in solid dark-blue with 95% confidence interval in green dot-dashed line of 8.40–14.60; pink line corresponds to regression line and orange lines to its confidence interval. Dashed gray lines at ±1.96 SD (−101.5 to 124.6 mg/dL) are the computed limits of agreement and solid light-blue lines are a 95% confidence interval.

**Figure 3 diagnostics-14-01906-f003:**
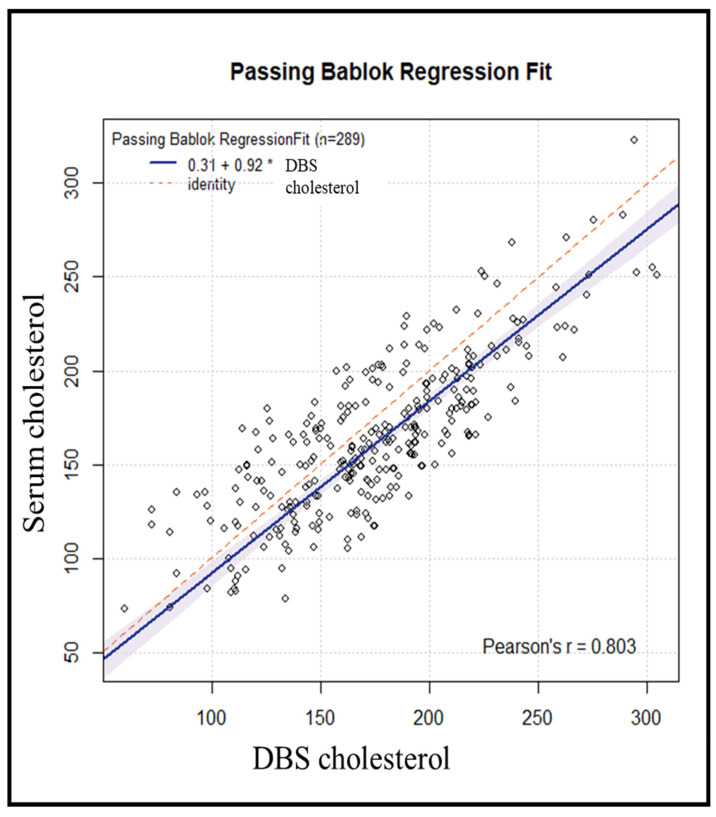
Passing–Bablok regression analysis of DBS cholesterol levels and serum cholesterol levels in 289 subjects (* = multiplication sign).

**Figure 4 diagnostics-14-01906-f004:**
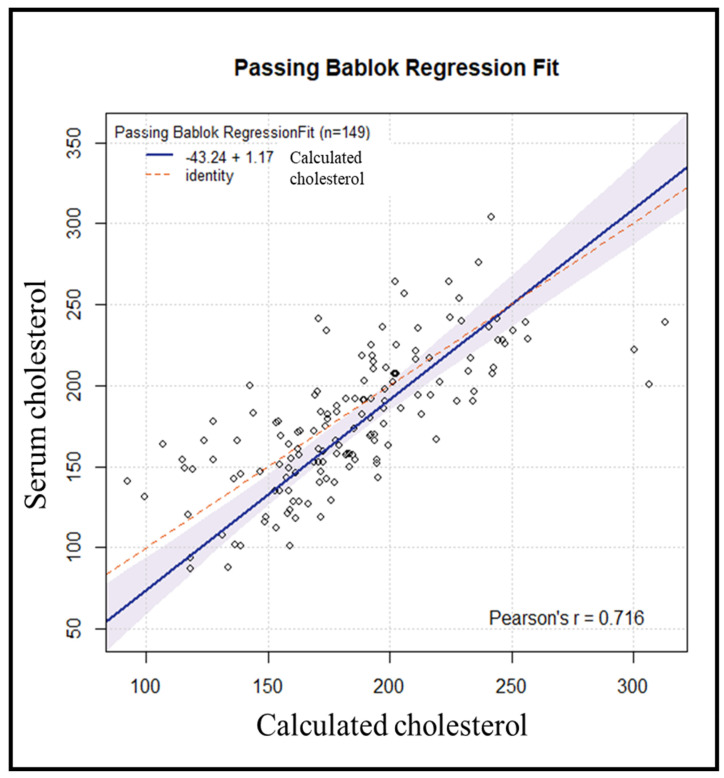
Passing−Bablok comparison between predicted values obtained from DBS and serum cholesterol.

**Figure 5 diagnostics-14-01906-f005:**
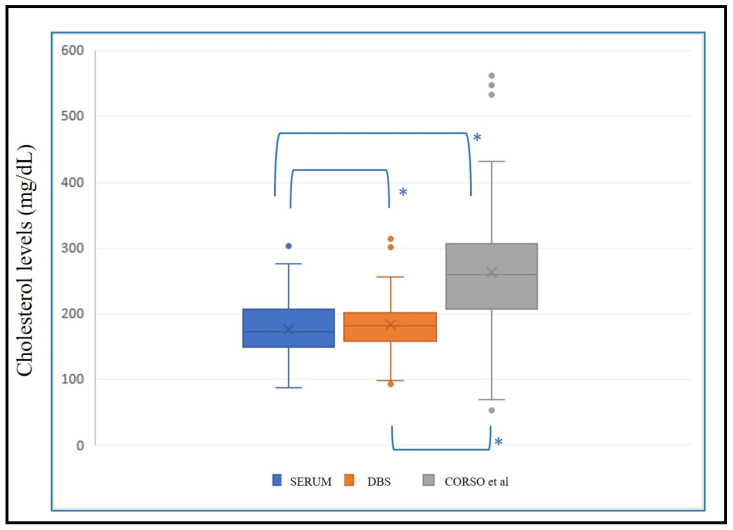
Box plot with levels of serum cholesterol, estimated cholesterol in DBS, and estimated cholesterol from external formula (Corso et al. [11]). *: *p* < 0.001.

**Table 1 diagnostics-14-01906-t001:** Within-day and between-day imprecision studies of DBS cholesterol measured in the DBS quality controls at three concentration levels (QC: quality control, CV: coefficient of variation, SD: standard deviation).

	SERUM (mg/dL)	DBS							
		WITHIN-DAY				BETWEEN-DAY (30 Days)			
		Mean (mg/dL)	SD (mg/dL)	CV (%)	n	Mean (mg/dL)	SD (mg/dL)	CV (%)	n
Low QC	149	136.35	13.83	10.14	25	145.66	20.52	14.09	27
Medium QC	220	228.25	20.31	8.90	23	223.83	23.66	10.57	27
High QC	283	275.00	22.30	8.11	26	256.27	34.06	13.29	27

**Table 2 diagnostics-14-01906-t002:** Comparison between values obtained by reference method in serum and manual enzymatic method in DBS. (IQR: Interquartile Range) * *p* = 0.001.

	SERUM (mg/dL)	DBS (mg/dL)	DBS-SERUM DIFFERENCES (mg/dL)
n	289	289	-
Mean	164	175.18 *	11.50
Median	162	173.96	14.89
Standard deviation	41	44.10	26.86
1st quartile	137	145.94	−11.02
3rdquartile	190	202.74	31.05
IQR	53	56.80	42.07
Maximum	323	304.75	58.04
Minimum	73	60.07	−54.78

## Data Availability

The original contributions presented in the study are included in the article; further inquiries can be directed to the corresponding author.

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
