# Peer review of "Total Cholesterol Determination Accuracy in Dried Blood Spots"

_diagnostics, 2024, doi:10.3390/diagnostics14171906_

Round 1

Reviewer 1 Report

Comments and Suggestions for Authors

Drs Bonet Estruch and colleagues present a comprehensive assessment of the agreement between levels of cholesterol measured in serum and those obtained from corresponding dried blood spot (DBS) samples. Analytical methods for extracting cholesterol from DBS are well described and show low relative variation, with CV~10% or less when performed on the same day. Other resultsbased on a mixture of descriptive summary measures, Bland-Altman plots, robust/resistant Passing-Bablok regression models, and a variety of non-parametric testsreveal that cholesterol extracted from DBS is slightly higher (~11 mg/dL, on average) than corresponding levels in the serum. The paper is well organized with graphs and figures to help convey their overall study design and ultimate conclusions.

There are some concerns about some of the tests performed and the potential that they may have ignored the inherent paired / linked nature of DBS and serum levels from the same patient. My comments are limited to statistical aspects of the paper and are given below; line numbers are approximate and relate to the submitted PDF proof copy.

Abstract, line~31: “… analysis in dot blood spots …” Should this be dried blood spots?

Statistical analysis

Lines ~146-147: The authors report that a “Mann-Whitney U test” was used to analyze their data. This test is designed for comparison of two separate (unrelated) samples. Comparing or testing for some sort of shift/discrepancy in the distributions of cholesterol from serum and DBS should involve a paired test of some kind—either a paired t-test, sign test, or signed-rank test—with the last two being distribution-free methods. Equivalently, these can be performed as a one-sample testing procedure (t or sign or signed-rank) applied to a set of computed differences (e.g., perform a test on the computed quantity DBS serum ).

Lines ~ 150-151: The MW U test is referenced here again, but it would seem to be another case where the information is paired in a non-ignorable way that needs to be taken into account. The n=149 validation samples are still linked/matched with the triple of serum, DBS, and calculated values based on Corso et al being derived each one of the 149 subjects.

Lines ~152-153: Please include a citation for the R (v.4.4.0) software program. It can be found by starting R and then typing citation() at the prompt. Additionally, the base or standard version of R does not have a routine for Passing-Bablok regression; that functionality must be added on using some additional package(s). Please specify which package was used (mcr ? deming ? ) and provide an appropriate citation for the package that was used.

Lines ~171-178: The authors describe how data were trimmed for the method comparison portion of the study and note they “… discard values exceeding the 90th percentile or falling below the 10th percentile of the difference between serum cholesterol and DBS …” This language (underlined for emphasis) implies that a set of differences (DBS serum) was first computed and then trimmed according to percentiles. However, the flow chart in Figure 1 shows trimming/exclusions done only on the side of the samples for DBS cholesterol and gives the impression that outlier removal from Tukey’s fence (excluded 14) and percentile trimming (exclude a further 72) was done for DBS and then the corresponding serum samples taken for the matching set of subjects. Could you please clarify in the text and possibly in Figure 1 which way the data were trimmed?

Lines ~179-184: The serum and DBS cholesterol measures are not separate independent samples, so testing each sample separately using the K-S test is not relevant. What is relevant is the behavior of the distribution of the differences (calculated as DBS serum). And even then, doing the K-S test (on the set of differences) won’t be particularly useful because of the extensive trimming that was done. Better to just declare from the outset that non-parametric / distribution-free methods were used for the analysis. In summarizing values for the DBS and serum cholesterol, the authors report a median ± SD. If you are reporting a median, then please provide the IQR instead; otherwise, if you report a SD as the measure of scale, then please report a mean as the measure of location. Given the amount of trimming that was done (leaving only the central 80% of the DBS distribution), the mean and median will be fairly similar so using the mean ± SD would be effective in this case. Is there a reason why serum values in Table 2 have less precision than the DBS values? Please use the same level of accuracy/precision when reporting data for both serum and DBS. Also, please expand Table 2 to include a third column summarizing the set of differences (DBS serum).

Lines ~187-189: The explanation of the Bland-Altman plot is too brief given what was actually drawn in Figure 2. Based on the figure, it appears that dashed gray lines set at ± 1.96 SD are the computed limits of agreement (LOA) and that solid blue surrounding lines are a 95% confidence interval (CI) for the LOA. It then appears the line type and color are reversed for the mean, with the solid blue line line representing the mean and dashed gray lines marking the 95% CI for the mean. Please make things consistent. There is no explanation for what the two rust/orange colored lines represent on the plot … either suppress those lines from the figure or explain what they measure. In the text (line 188-189), the range for the LOA should be 101.5 to +124.6 (not 101.5 to 124.6).

Please provide the 95% CI for the mean difference. We are told and shown in the figure that the bias is that DBS cholesterol is, on average, 11.5 mg/dL greater than serum cholesterol from the same sample, but we have no confidence interval for this mean difference and no way to know whether the shift is significant.

Please also calculate the Pearson correlation coefficient between the averages and the differences that are used to make the Bland-Altman plot and test whether that correlation significantly differs from 0. This is referred to as Pitman’s test and addresses whether the two distributions (cholesterol level in DBS and serum) have the same standard deviation. The earlier comment (concerning a test and confidence interval for the mean difference) addresses whether the two distributions have the same typical response and together with Pitman’s test will then explain whether the scale/dispersion is similar or different as well.

Lines 190-194: The authors give an ROC curve with summarizing information and describe that “… sensitivity was 100%, identifying all positive cases, ….” Yet the notion of “case” is never defined. My suspicion is that they assigned a numeric code of 1 to serum samples and a code of 0 to DBS and then did the ROC analysis on the n=289*2=578 observations from this artificial double-length date set. If this is the case, then that would be inappropriate because (again), there is only one sample of 289 observations with paired DBS / serum measurements made for each sample. My suggestion is to strike this paragraph and the corresponding Figure 3 as it doesn’t really add to anything.

Line 199: What is the “accuracy” value that is cited as being equal to 0.9604?

Lines 200-201: Most software packages ( mcr in R, for instance ) also provide 95% CIs for the estimated intercept and slope of the fitted line. This would be crucial information as one would be especially interested in whether the slope significantly differed from 1 and whether the intercept significantly differed from 0. Please review the output for Figure 4 and include the relevant information in the text. E.g., “The Passing-Bablok regression yielded an estimated intercept of 0.31 (95% CI: xx—yy) with corresponding estimated slope of 0.92 (95% CI: rr—ss).” You don’t have to repeat the equation of the line and the correlation coefficient (line ~201) since that information is given expressly in the legend for Figure 4.

Lines 205-207: Please report the intercept and slope in the text with supporting 95% CIs so the reader can determine whether those parameters differ significantly from 0 and 1, respectively. Please also suppress the equation and Pearson correlation coefficient from the text (line 207) since it is given in the Figure 5.

Line 208: It’s fine to give the regression line from Corso et al here, but please report it in the text with the intercept first and slope second (e.g., DBS = 64.86 + 0.5217*Serum) to be consistent with how the results are given in the two figures. At the same time I would ask the authors to carefully check their own work concerning what is estimated from the regression line. In the Pa-Ba regression models they fit, the serum cholesterol is the response (y) and the DBS is the explanatory (x) variable. In Corso, the roles are reversed and Corso’s line predicts DBS conditional upon knowing the serum value. [For what it’s worth: the manner used in Corso is correct in that the inexpensive / test variable (here DBS) is usually selected as the response (y) and the more expensive gold-standard / reference variable (here serum) is chosen as the explanatory (x) variable.]

Lines 210-216: In this paragraph the authors describe having serum cholesterol (measured), estimated DBS from their model, and estimated DBS from Corso’s equation, with these last two estimated quantities based on the same n=149 serum values. Here again, all three values are linked/matched according to the particular specimen. So K-S tests done separately don’t really matter and the necessary tests should be some sort of signed-rank or sign test being applied to each potential set of differences ( DBS serum , Corso serum, Corso DBS ).

Lines 261-262: The explanation of a smaller DBS vs serum difference (11 mg/dL vs 30 mg/dL reported from others) may also be from the more elaborate trimming of data used by the current authors (e.g., removal of observations beyond Tukey’s upper/lower fence; then excluding the top/bottom 10% of what remains) if the other investigators didn’t trim or exclude observations to the same degree.

Author Response

Thanks for your comments and suggestions.

We have made the following changes:

Comment 1: Abstract, line~31: “… analysis in dot blood spots …” Should this be dried blood spots?

Response 1: Agree and changed.

Comment 2: “Lines ~146-147: The authors report that a “Mann-Whitney U test” was used to analyze their data. This test is designed for comparison of two separate (unrelated) samples. Comparing or testing for some sort of shift/discrepancy in the distributions of cholesterol from serum and DBS should involve a paired test of some kind—either a paired t-test, sign test, or signed-rank test—with the last two being distribution-free methods. Equivalently, these can be performed as a one-sample testing procedure (t or sign or signed-rank) applied to a set of computed differences (e.g., perform a test on the computed quantity DBS − serum).

Response 2:  We understand your vision and suggestions, but although we have determined the same biomarker from the same extraction day and same patient, samples are different (serum vs. DBS from EDTA tube) which includes different matrix, the extraction method and sample elaboration also differ (total serum vs. methanol extraction) and finally, the technique used to determine the concentration (autoanalyzer vs. manual technique). Due to all these details and after discussing this point with the statistical staff, we think that the Mann-Whitney test is the best statistic for comparing both techniques as they are unrelated and non parametrics.

Comment 3: Lines ~ 150-151: The MW U test is referenced here again, but it would seem to be another case where the information is paired in a non-ignorable way that needs to be taken into account. The n=149 validation samples are still linked/matched with the triple of serum, DBS, and calculated values based on Corso et al being derived each one of the 149 subjects.

Response 3: As mention above, even though we analyzed the same biomarker, the result found in serum cannot be related to the result found in DBS due to different matrix, processing and analytical method.

Comment 4: Lines ~152-153: Please include a citation for the R (v.4.4.0) software program. It can be found by starting R and then typing citation() at the prompt. Additionally, the base or standard version of R does not have a routine for Passing-Bablok regression; that functionality must be added on using some additional package(s). Please specify which package was used (mcr ? deming ? ) and provide an appropriate citation for the package that was used.

Response 4:  Agree and changed as followed (line 160): “method comparison was accomplished by the R Statistical Software (v4.4.0; R Core Team 2024). Bland-Altman plot was obtained from Bland-AltmanLeh:Plots (Slightly Extended) R package (v0.3.1; 2015) and Passing Bablok plot from mcr (Method Comparison Regression) R package (v1.3.3; 2023).”

Comment 5: “lines ~171-178: The authors describe how data were trimmed for the method comparison portion of the study and note they “… discard values exceeding the 90th percentile or falling below the 10th percentile of the difference between serum cholesterol and DBS …” This language (underlined for emphasis) implies that a set of differences (DBS − serum) was first computed and then trimmed according to percentiles. However, the flow chart in Figure 1 shows trimming/exclusions done only on the side of the samples for DBS cholesterol and gives the impression that outlier removal from Tukey’s fence (excluded 14) and percentile trimming (exclude a further 72) was done for DBS and then the corresponding serum samples taken for the matching set of subjects. Could you please clarify in the text and possibly in Figure 1 which way the data were trimmed?”

Response 5: Thanks for your comment. We have clarified both the text and the figure 1. Line 185: “Furthermore, to conduct method comparison analysis, we decided to calculate the difference between serum cholesterol and DBS and discard values exceeding the 90th percentile or falling below the 10th percentile of the difference”.

Comment 6: “Lines ~179-184: The serum and DBS cholesterol measures are not separate independent samples, so testing each sample separately using the K-S test is not relevant. What is relevant is the behavior of the distribution of the differences (calculated as DBS − serum). And even then, doing the K-S test (on the set of differences) won’t be particularly useful because of the extensive trimming that was done. Better to just declare from the outset that non-parametric / distribution-free methods were used for the analysis. In summarizing values for the DBS and serum cholesterol, the authors report a median ± SD. If you are reporting a median, then please provide the IQR instead; otherwise, if you report a SD as the measure of scale, then please report a mean as the measure of location. Given the amount of trimming that was done (leaving only the central 80% of the DBS distribution), the mean and median will be fairly similar so using the mean ± SD would be effective in this case. Is there a reason why serum values in Table 2 have less precision than the DBS values? Please use the same level of accuracy/precision when reporting data for both serum and DBS. Also, please expand Table 2 to include a third column summarizing the set of differences (DBS − serum).

Response 6: As you have suggested we have deleted the sentence regarding to K-S test. We have also changed median for mean. The reason why serum values show less precision is because we have not matched this part as we have use cholesterol values as gold standard technique and only adjust DBS values as far as possible. We have added the third column suggested in table 2.

Comment 7: “Lines ~187-189: The explanation of the Bland-Altman plot is too brief given what was actually drawn in Figure 2. Based on the figure, it appears that dashed gray lines set at ± 1.96 SD are the computed limits of agreement (LOA) and that solid blue surrounding lines are a 95% confidence interval (CI) for the LOA. It then appears the line type and color are reversed for the mean, with the solid blue line representing the mean and dashed gray lines marking the 95% CI for the mean. Please make things consistent. There is no explanation for what the two rust/orange colored lines represent on the plot … either suppress those lines from the figure or explain what they measure. In the text (line 188-189), the range for the LOA should be −101.5 to +124.6 (not 101.5 to 124.6).

Please provide the 95% CI for the mean difference. We are told and shown in the figure that the bias is that DBS cholesterol is, on average, 11.5 mg/dL greater than serum cholesterol from the same sample, but we have no confidence interval for this mean difference and no way to know whether the shift is significant.

Please also calculate the Pearson correlation coefficient between the averages and the differences that are used to make the Bland-Altman plot and test whether that correlation significantly differs from 0. This is referred to as Pitman’s test and addresses whether the two distributions (cholesterol level in DBS and serum) have the same standard deviation. The earlier comment (concerning a test and confidence interval for the mean difference) addresses whether the two distributions have the same typical response and together with Pitman’s test will then explain whether the scale/dispersion is similar or different as well.

Response 7: Proper explanation has been added at the text and the figure 2 according to the results of Bland Altman plot. Slightly differences in color may be found to make the difference between mean and 1.96 SD. If any other information or data is needed, it could be added in the future. 95% CI for the mean difference has been added and also, regression data and F-Snedecor test results have been added. Line 197: Bland-Altman plot (Fig. 2) was used to evaluate agreement between DBS and serum cholesterol determination, revealing an average bias of 11.5 mg/dL (dark solid blue line) with 95% confidence interval in green dot-dashed line of 11.24-17.90 mg/dL. The computed limits of agreement corresponding to ± 1.96 SD is from -101,5 to 124,6 mg/dL (dashed gray lines) with a 95% confidence interval (light solid blue lines). The Pearson correlation coefficient between the averages and the differences is 0.80 (95% CI: 0.76 - 0.84). The F-Snedecor test used with unrelated samples gives a result of P = 0.4519 which means that the two distributions have the same typical response and the dispersion is similar.

Comment 8: “Lines 190-194: The authors give an ROC curve with summarizing information and describe that “…sensitivity was 100%, identifying all positive cases, ….” Yet the notion of “case” is never defined. My suspicion is that they assigned a numeric code of 1 to serum samples and a code of 0 to DBS and then did the ROC analysis on the n=289*2=578 observations from this artificial double-length date set. If this is the case, then that would be inappropriate because (again), there is only one sample of 289 observations with paired DBS / serum measurements made for each sample. My suggestion is to strike this paragraph and the corresponding Figure 3 as it doesn’t really add to anything.

Response 8: We have strike the paragraph about ROC curve. We think that this information will be useful in further studies.

Comments 9: “Line 199: What is the “accuracy” value that is cited as being equal to 0.9604?

Response 9: Thanks for your suggestion. We have added the following sentence (line 210): “The bias correction factor (Cb) value that is a measure of accuracy was 0.9604, suggesting high accuracy of the DBS method”.

Comments 10: Lines 200-201: Most software packages ( mcr in R, for instance ) also provide 95% CIs for the estimated intercept and slope of the fitted line. This would be crucial information as one would be especially interested in whether the slope significantly differed from 1 and whether the intercept significantly differed from 0. Please review the output for Figure 4 and include the relevant information in the text. E.g., “The Passing-Bablok regression yielded an estimated intercept of 0.31 (95% CI: xx—yy) with corresponding estimated slope of 0.92 (95% CI: rr—ss).” You don’t have to repeat the equation of the line and the correlation coefficient (line ~201) since that information is given expressly in the legend for Figure 4.

Response 10: We have changed the text as follows (line 211): “The Passing-Bablok regression yielded an estimated intercept of 0.31 (95% CI: -13.44 - +12.41) with corresponding estimated slope of 0.92 (95% CI: 0.85-0.99)”.

Comment 11: “Lines 205-207: Please report the intercept and slope in the text with supporting 95% CIs so the reader can determine whether those parameters differ significantly from 0 and 1, respectively. Please also suppress the equation and Pearson correlation coefficient from the text (line 207) since it is given in the Figure 5”.

Response 11: We have changed the text as follows (line 218): Comparing the results of serum cholesterol versus calculated cholesterol yielded a Passing-Bablok regression line with an estimated intercept of -43.24 [95% CI: -74.54 - (-16.68)] with corresponding estimated slope of 1.17 (95% CI: 1.03 - 1.35) (Fig. 4).

Comment 12: Line 208: It’s fine to give the regression line from Corso et al here, but please report it in the text with the intercept first and slope second (e.g., DBS = 64.86 + 0.5217*Serum) to be consistent with how the results are given in the two figures. At the same time I would ask the authors to carefully check their own work concerning what is estimated from the regression line. In the Pa-Ba regression models they fit, the serum cholesterol is the response (y) and the DBS is the explanatory (x) variable. In Corso, the roles are reversed and Corso’s line predicts DBS conditional upon knowing the serum value. [For what it’s worth: the manner used in Corso is correct in that the inexpensive / test variable (here DBS) is usually selected as the response (y) and the more expensive gold-standard / reference variable (here serum) is chosen as the explanatory (x) variable.]

Response 12: Thanks for your suggestion. We had noticed that Corso`s roles are reversed but our purpose is to predict cholesterol from DBS, therefore with the regression line we can use DBS (x) to obtain predicted cholesterol serum (y), being DBS considered as the reference method in this case.

Comment 13: Lines 210-216: In this paragraph the authors describe having serum cholesterol (measured), estimated DBS from their model, and estimated DBS from Corso’s equation, with these last two estimated quantities based on the same n=149 serum values. Here again, all three values are linked/matched according to the particular specimen. So K-S tests done separately don’t really matter and the necessary tests should be some sort of signed-rank or sign test being applied to each potential set of differences (DBS − serum, Corso − serum, Corso − DBS).

Response 13: As mention above, even though we analyzed the same biomarker, the result found in serum cannot be related to the result found in DBS due to different matrix, processing and analytical method. So, we used MannWhitney test to compare serum with DBS and Corso, but Wilcoxon test to compare DBS and Corso because we consider these samples related as they are the kind of sample.

Line 158 “Wilcoxon signed-rank test was used to compare differences in external validation.”

Line 229 “No significant differences between DBS and Corso were observed either (p<0.001) by Wilcoxon test”

Comment 14: Lines 261-262: The explanation of a smaller DBS vs serum difference (11 mg/dL vs 30 mg/dL reported from others) may also be from the more elaborate trimming of data used by the current authors (e.g., removal of observations beyond Tukey’s upper/lower fence; then excluding the top/bottom 10% of what remains) if the other investigators didn’t trim or exclude observations to the same degree.

Response 14: Thanks for your vision. Without trimming the data, we found only a difference of 14 mg/dL, but the predicted formula obtained after trimming was better so we decided this proceed was more suitable for its future use.

Reviewer 2 Report

Comments and Suggestions for Authors

I had the pleasure of reviewing the paper by Estruch et al., which investigates the diagnostic accuracy of measuring total cholesterol in dried blood spots (DBS) using Whatman 903 paper to identify familial hypercholesterolemia. The results demonstrate strong precision and accuracy, with an area under the curve of 0.976, indicating that the DBS method is a reliable approach for neonatal screening of familial hypercholesterolemia.

Comments: This is a very well-written paper with important results. I have only a few minor suggestions:

  1. Definition of "Level 1 Genomic Alteration": In lines 55-56, I recommend defining "Level 1 genomic alteration" to make the introduction more accessible to a broader audience.

  2. Inclusion Criteria and Newborns: The authors mentioned that the inclusion criteria were patients over 18 years old. I wonder how this might differ for newborns, considering the project's intended expansion. Do the authors foresee any challenges with this different cohort?

  3. Figure Quality: The sharpness of Figure 1 could be improved. I suggest uploading a higher resolution version.

  4. Decimal Consistency: In line 158, I recommend standardizing all numerical values to one or two decimal places for consistency.

Good luck!

Author Response

Thanks for your comments and suggestions.

We have made the following changes:

Comment 1: Definition of "Level 1 Genomic Alteration": In lines 55-56, I recommend defining "Level 1 genomic alteration" to make the introduction more accessible to a broader audience.

Response 1: We agree with this point. Therefore, we have explain this part in the text in lines 56-60: “Level 1 genomic alterations are defined by Centers for Disease Control and Prevention Office of Public Health Genomics as those that can be detected by genetic testing and their detection can have a positive impact on public health through early interventions to reduce their morbidity and mortality. FH is considered one of these level 1 genomic alterations”

Comment 2: Inclusion Criteria and Newborns: The authors mentioned that the inclusion criteria were patients over 18 years old. I wonder how this might differ for newborns, considering the project's intended expansion. Do the authors foresee any challenges with this different cohort?

Response 2: Thank you for pointing this out. Selected patients were over 18 years old because our main purposed was proving that DBS cholesterol has high reproducibility and accuracy. Our group have been working in this field during few years and we have newborn samples but this data have not been published yet. In future papers, our aim will be to demonstrate that this technique can be used regardless of age.

Comment 3: Figure Quality: The sharpness of Figure 1 could be improved. I suggest uploading a higher resolution version.

Response 3: Agree and changed.

Comment 4: Decimal Consistency: In line 158, I recommend standardizing all numerical values to one or two decimal places for consistency.

Response 4: Agree and changed

Round 2

Reviewer 1 Report

Comments and Suggestions for Authors

Thank you for your attention to the comments. The manuscript is, I think, much improved.

I still have reservations about some of the tests/analyses that are more fully described in the comments below.

Comment 1: Thank you for fixing.

Comment 2: While I can appreciate that serum and DBS may have been treated differently for all the reasons described (different matrix, separate extraction methods, different techniques [auto-analyzer vs manual], etc.), the main point is that each pair of serum and DBS was measured on the same patient. Whether samples are correlated/paired depends on the design, and not how the data appears or how things are treated after the fact. This pairing is essential for things like the Bland-Altman plot and Passing-Bablok regression, where the unique and specific linkage between serum and DBS for each patient must be preserved for either of these analyses to make sense. By contrast, analyzing things with a Mann-Whitney U test and treating the samples as unrelated assumes there is no linkage and makes it impossible to do things like Bland-Altman or any kind of regression. It is not possible to hold these two views at the same time: either the samples are paired and linked — enabling Bland-Altman and Pa-Ba regressions, but then requiring signed-rank or paired t-tests — or the samples are independent (allowing Mann-Whitney U, but then not allowing Bland-Altman or Pa-Ba regression).

Comment 3: This is still a case where DBS and serum are still paired and the analysis should reflect that.

Comment 4: Thank you for adding the references to the relevant packages.

Comment 5: Thank you for the explanation in the text and the revision to Figure 1. Things are much clearer now.

Comment 6: Thank you for including the additional column in Table 2. Thank you also for updating the sentence to use the mean +/- SD. Thank you for the explanation concerning level of precision in reporting the values in Table 2. The mean difference (average of the differences) should equal 175.18-164 ~ 11.2 (not 10.82). The same difference (11.2) should be what the average bias is in the Bland-Altman plot. Because of way a confidence intervals for the mean difference is constructed, the limits (reported as 11.24 to 17.90 mg/dL) should be symmetric and average to 11.2 … but they don’t and instead average to 14.57 mg/dL, which is different from anything reported before. Please check and revise. (Based on the mean difference of 11.2 and reported SD of the differences, the 95% CI for the average bias is 8.1 to 14.3 mg/dL).

Comment 7: This is a very nice explanation of the elements shown in the plot. One issue that remains is use of the usual F-test for equality of variances. That test presumes two independent/unrelated samples, but in fact we have just one sample that is paired (as evidenced by the ability to make Bland-Altman plots and speak about the set of differences that result from the unique way in which DBS and serum are linked to the same patient). So Snedecor’s F-test is not appropriate and the correlation coefficient that is reported (0.80 [95% CI: 0.76–0.84]) implies a significant association between the two, which is equivalent to finding the two measures have different scale. [Pitman EJG. (1939). A note on normal correlation. Biometrika, 31:9–12.]. The F-test results and final statement (“… which means that the two distributions have … dispersion is similar.”) should be updated. The average difference (based on the visual of the 95% CI on Figure 2 being offset from 0) is shifted away from 0 and also the Pa-Ba regression has a slope coefficient that significantly differs from 1 (the 95% CI is 0.85 to 0.99). These point to there being a proportional difference in the mean responses from DBS and serum as well as the two measures having different levels of scale ( 41 vs 44.1 mg/dL).

Comment 8: Thank you for deleting.

Comment 9: Thank you for clarifying. That bias correction factor (Cb ) may be unfamiliar to readers and is discussed in Lin L-IK. (1989). A concordance correlation coefficient to evaluate reproducibility. Biometrics, 45:255-268. Perhaps this should be added to the reference?

Comments 10/11: Thank you for including the confidence intervals for the associated Pa-Ba regression lines.

Comment 12: Thank you for fixing.

Comment 13: I still maintain that all these measured are paired in some meaningful way that should be reflected in the analysis. I think (line 229 of the revised MS) that if the p-value is <0.001, then it should be “A significant difference (p<0.001) between DBS and Corso was noted.” Also just noticing now that the footnote mark for Corso is reported as #10, which instead points to Held et al in the reference list. Corso et al is reference #11.

Comment 14: Thank you for checking that differences are still “small” even without trimming

Author Response

Thanks for your answer and suggestions.

We proceed to explain the part that have been corrected in this new version: 

Comment 2: “While I can appreciate that serum and DBS may have been treated differently for all the reasons described (different matrix, separate extraction methods, different techniques [auto-analyzer vs manual], etc.), the main point is that each pair of serum and DBS was measured on the same patient. Whether samples are correlated/paired depends on the design, and not how the data appears or how things are treated after the fact. This pairing is essential for things like the Bland-Altman plot and Passing-Bablok regression, where the unique and specific linkage between serum and DBS for each patient must be preserved for either of these analyses to make sense. By contrast, analyzing things with a Mann-Whitney U test and treating the samples as unrelated assumes there is no linkage and makes it impossible to do things like Bland-Altman or any kind of regression. It is not possible to hold these two views at the same time: either the samples are paired and linked — enabling Bland-Altman and Pa-Ba regressions, but then requiring signed-rank or paired t-tests — or the samples are independent (allowing Mann-Whitney U, but then not allowing Bland-Altman or Pa-Ba regression).”

Response 2: We have calculated the Wilcoxon test for both regression analysis: comparison regression and internal validation of the formula, and it have been changed in the text. Lines 154 and 158.

Comment 3: “This is still a case where DBS and serum are still paired and the analysis should reflect that.”

Response 3: Explained in response 2.

Comment 6: “Thank you for including the additional column in Table 2. Thank you also for updating the sentence to use the mean +/- SD. Thank you for the explanation concerning level of precision in reporting the values in Table 2. The mean difference (average of the differences) should equal 175.18-164 ~ 11.2 (not 10.82). The same difference (11.2) should be what the average bias is in the Bland-Altman plot. Because of way a confidence intervals for the mean difference is constructed, the limits (reported as 11.24 to 17.90 mg/dL) should be symmetric and average to 11.2 … but they don’t and instead average to 14.57 mg/dL, which is different from anything reported before. Please check and revise. (Based on the mean difference of 11.2 and reported SD of the differences, the 95% CI for the average bias is 8.1 to 14.3 mg/dL).”

Response 6: The mean difference has been corrected, as you have commented, it is the same as Bland Altman results, it was due to a lack in decimal numbers. Confidence intervals has also been checked. Line 199: “with 95% confidence interval in green dot-dashed line of 8.40-14.60 mg/dL.”

Comment 7: “This is a very nice explanation of the elements shown in the plot. One issue that remains is use of the usual F-test for equality of variances. That test presumes two independent/unrelated samples, but in fact we have just one sample that is paired (as evidenced by the ability to make Bland-Altman plots and speak about the set of differences that result from the unique way in which DBS and serum are linked to the same patient). So Snedecor’s F-test is not appropriate and the correlation coefficient that is reported (0.80 [95% CI: 0.76–0.84]) implies a significant association between the two, which is equivalent to finding the two measures have different scale. [Pitman EJG. (1939). A note on normal correlation. Biometrika, 31:9–12.]. The F-test results and final statement (“… which means that the two distributions have … dispersion is similar.”) should be updated. The average difference (based on the visual of the 95% CI on Figure 2 being offset from 0) is shifted away from 0 and also the Pa-Ba regression has a slope coefficient that significantly differs from 1 (the 95% CI is 0.85 to 0.99). These point to there being a proportional difference in the mean responses from DBS and serum as well as the two measures having different levels of scale ( 41 vs 44.1 mg/dL).”

Response 7: As both method have been considered as related samples, we have changed Snedecor test and calculated Pitman’s test. Line 203.

Comment 9: “Thank you for clarifying. That bias correction factor (Cb ) may be unfamiliar to readers and is discussed in Lin L-IK. (1989). A concordance correlation coefficient to evaluate reproducibility. Biometrics, 45:255-268. Perhaps this should be added to the reference?”

Response 9: The suggested reference has been added in the bibliography. Line 209 and 359.

Comment 13: “I still maintain that all these measured are paired in some meaningful way that should be reflected in the analysis. I think (line 229 of the revised MS) that if the p-value is <0.001, then it should be “A significant difference (p<0.001) between DBS and Corso was noted.” Also just noticing now that the footnote mark for Corso is reported as #10, which instead points to Held et al in the reference list. Corso et al is reference #11.”

Response 13:  Statistics have been changed for paired samples as commented in response 2. The sentence suggested is better than the one before and it has been changed. Reference numbers have been changed due to the additional citation.  

As you can observe Wilcoxon test show significant differences between serum and DBS, but as it can be detected in the figure 5, Corso’s formula shows a higher level of predicted cholesterol values and also a higher dispersion, so to emphasize that our formula, though not given the same result, can give a better positive and negative predicted value and better specificity, these data have been added (line 225-235): “In addition, using a total cholesterol cutoff value of 190 mg/dL, we observed that in DBS 76 of 91 cholesterol predicted values of validation test were below 190 mg/dL and in accordance with the corresponding serum cholesterol values, and 51 of 58 were above 190 mg/dL in accordance with the serum values. Nevertheless, using Corso’s formula, we obtained 23 of 91 cholesterol values below 190 mg/dL and in accordance with the corresponding serum cholesterol values, and 57 of 58 above 190 mg/dL.

 Applying our regression formula, a sensitivity value of 0.88 and a specificity value of 0.84, with a positive predictive value of 0.77 and a negative predictive value of 0.92 was found, whereas with Corso’s formula, it has been obtained a sensitivity of 0.98 and specificity of 0.25, with a positive predictive value of 0.46 and a negative predictive value of 0.96.” 

Round 3

Reviewer 1 Report

Comments and Suggestions for Authors

Thank you for the updates and revisions. 

I think the addition of sensitivity and specificity estimates using a pre-defined threshold of 190 mg/dL is a useful addition.

In regards to Pitman's test (line ~197--200) has an issue with the reported p-value of 0.4519. Authors have already reported the Pearson correlation between the mean and the difference is 0.80 (95% CI: 0.76--0.84). Because this 95% confidence interval fails to include a correlation value of 0, we know that a formal test of r_p = 0 would have a p-value less than 0.05. So reporting p=0.4519 (non-significant) is an error. It's pretty clear (from the [lower limit of the] CI and the estimate of 0.80) that the correlation is very far from 0 (i.e., significantly differs from 0), so p < 0.001 and the conclusion is that the two measures (DBS and serum) actually *do* have different levels of dispersion.

Author Response

Thanks for your revision.

Comment: “In regards to Pitman's test (line ~197--200) has an issue with the reported p-value of 0.4519. Authors have already reported the Pearson correlation between the mean and the difference is 0.80 (95% CI: 0.76--0.84). Because this 95% confidence interval fails to include a correlation value of 0, we know that a formal test of r_p = 0 would have a p-value less than 0.05. So reporting p=0.4519 (non-significant) is an error. It's pretty clear (from the [lower limit of the] CI and the estimate of 0.80) that the correlation is very far from 0 (i.e., significantly differs from 0), so p < 0.001 and the conclusion is that the two measures (DBS and serum) actually *do* have different levels of dispersion.”

Response: As you have pointed out, this was a mistake. We have checked the data and our results, and we have done the Pitman’s test again having a result of p= 0.0008, so, it has been corrected in the text as follows: (Line 200)“The Pearson correlation coefficient between the averages and the differences is 0.80 (95% CI: 0.76 - 0.84). Pitman's test used with related samples gives a result of p < 0.001 which means that the two distributions have different levels of dispersion.”